# Balanced Meta-Softmax
# for Long-Tailed Visual Recognition

**Jiawei Ren[1], Cunjun Yu[1], Shunan Sheng[1,2], Xiao Ma[1,3],**
**Haiyu Zhao[1*], Shuai Yi[1], Hongsheng Li[4]**
[1] SenseTime Research
[2] Nanyang Technological University
[3] National University of Singapore
[4] Multimedia Laboratory, The Chinese University of Hong Kong
{renjiawei, zhaohaiyu, yishuai}@sensetime.com    cunjun.yu@gmail.com
shen0152@e.ntu.edu.sg    xiao-ma@comp.nus.edu.sg    hsli@ee.cuhk.edu.hk

## Abstract

Deep classifiers have achieved great success in visual recognition. However, real-world data is long-tailed by nature, leading to the mismatch between training and testing distributions. In this paper, we show that the Softmax function, though used in most classification tasks, gives a biased gradient estimation under the long-tailed setup. This paper presents Balanced Softmax, an elegant unbiased extension of Softmax, to accommodate the label distribution shift between training and testing. Theoretically, we derive the generalization bound for multiclass Softmax regression and show our loss minimizes the bound. In addition, we introduce Balanced Meta-Softmax, applying a complementary Meta Sampler to estimate the optimal class sample rate and further improve long-tailed learning. In our experiments, we demonstrate that Balanced Meta-Softmax outperforms state-of-the-art long-tailed classification solutions on both visual recognition and instance segmentation tasks.[†]

## 1 Introduction

Most real-world data comes with a long-tailed nature: a few high-frequency classes (or head classes) contributes to most of the observations, while a large number of low-frequency classes (or tail classes) are under-represented in data. Taking an instance segmentation dataset, LVIS [7], for example, the number of instances in *banana* class can be thousands of times more than that of a *bait* class. In practice, the number of samples per class generally decreases from head to tail classes exponentially. Under the power law, the tails can be undesirably heavy. A model that minimizes empirical risk on long-tailed training datasets often underperforms on a class-balanced test dataset. As datasets are scaling up nowadays, the long-tailed nature poses critical difficulties to many vision tasks, e.g., visual recognition and instance segmentation.

An intuitive solution to long-tailed task is to re-balance the data distribution. Most state-of-the-art (SOTA) methods use the class-balanced sampling or loss re-weighting to "simulate" a balanced training set [3, 31]. However, they may under-represent the head class or have gradient issues during optimization. Cao et al. [4] introduced Label-Distribution-Aware Margin Loss (LDAM), from the perspective of the generalization error bound. Given fewer training samples, a tail class should have a higher generalization error bound during optimization. Nevertheless, LDAM is derived from the hinge loss, under a binary classification setup and is not suitable for multi-class classification.

---

[*]Corresponding author
[†]Code available at https://github.com/jiawei-ren/BalancedMetaSoftmax

We propose *Balanced Meta-Softmax* (BALMS) for long-tailed visual recognition. We first show that the Softmax function is intrinsically biased under the long-tailed scenario. We derive a Balanced Softmax function from the probabilistic perspective that explicitly models the test-time label distribution shift. Theoretically, we found that optimizing for the Balanced Softmax cross-entropy loss is equivalent to minimizing the generalization error bound. Balanced Softmax generally improves long-tailed classification performance on datasets with moderate imbalance ratios, e.g., CIFAR-10-LT [18] with a maximum imbalance factor of 200. However, for datasets with an extremely large imbalance factor, e.g., LVIS [7] with an imbalance factor of 26,148, the optimization process becomes difficult. Complementary to the loss function, we introduce the *Meta Sampler*, which learns to re-sample for achieving high validation accuracy by meta-learning. The combination of Balanced Softmax and Meta Sampler could efficiently address long-tailed classification tasks with high imbalance factors.

We evaluate BALMS on both long-tailed image classification and instance segmentation on five commonly used datasets: CIFAR-10-LT [18], CIFAR-100-LT [18], ImageNet-LT [23], Places-LT [34] and LVIS [7]. On all datasets, BALMS outperforms state-of-the-art methods. In particular, BALMS outperforms all SOTA methods on LVIS, with an extremely high imbalanced factor, by a large margin.

We summarize our contributions as follows: 1) we theoretically analyze the incapability of Softmax function in long-tailed tasks; 2) we introduce Balanced Softmax function that explicitly considers the label distribution shift during optimization; 3) we present Meta Sampler, a meta-learning based re-sampling strategy for long-tailed learning.

## 2 Related Works

**Data Re-Balancing.** Pioneer works focus on re-balancing during training. Specifically, re-sampling strategies [19, 5, 8, 10, 26, 2, 1] try to restore the true distributions from the imbalanced training data. Re-weighting, i.e., cost-sensitive learning [31, 11, 12, 24], assigns a cost weight to the loss of each class. However, it is argued that over-sampling inherently overfits the tail classes and under-sampling under-represents head classes' rich variations. Meanwhile, re-weighting tends to cause unstable training especially when the class imbalance is severe because there would be abnormally large gradients when the weights are very large.

**Loss Function Engineering.** Tan et al. [30] point out that randomly dropping some scores of tail classes in the Softmax function can effectively help, by balancing the positive gradients and negative gradients flowing through the score outputs. Cao et al. [4] show that the generalization error bound could be minimized by increasing the margins of tail classes. Hayat et al. [9] modify the loss function based on Bayesian uncertainty. Li et al. [20] propose two novel loss functions to balance the gradient flow. Khan et al. [17] jointly learn the model parameters and the class-dependent loss function parameters. Ye et al. [32] force a large margin for minority classes to prevent feature deviation. We progress this line of works by introducing probabilistic insights that also bring empirical improvements. We show in this paper that an ideal loss function should be unbiased under the long-tailed scenarios.

**Meta-Learning.** Many approaches [13, 25, 27] have been proposed to tackle the long-tailed issue with meta-learning. Many of them [13, 25] focus on optimizing the weight-per-sample as a learnable parameter, which appears as a hyper-parameter in the sample-based re-weight approach. This group of methods requires a clean and unbiased dataset as a meta set, i.e., development set, which is usually a fixed subset of the training images and use bi-level optimization to estimate the weight parameter.

**Decoupled Training.** Kang et al. [16] point out that decoupled training, a simple yet effective solution, could significantly improve the generalization issue on long-tailed datasets. The classifier is the only under-performed component when training in imbalanced datasets. However, in our experiments, we found this technique is not adequate for datasets with extremely high imbalance factors, e.g., LVIS [7]. Interestingly in our experiments, we observed that decoupled training is complementary to our proposed BALMS, and combining them results in additional improvements.

## 3 Balanced Meta-Softmax

The major challenge for long-tailed visual recognition is the mismatch between the imbalanced training data distribution and the balanced metrics, e.g., mean Average Precision (mAP), that encourage

minimizing error on a balanced test set. Let $\mathcal{X} = \{x_i, y_i\}, i \in \{1, \cdots, n\}$ be the balanced test set, where $x_i$ denotes a data point and $y_i$ denotes its label. Let $k$ be the number of classes, $n_j$ be the number of samples in class $j$, where $\sum_{j=1}^{k} n_j = n$. Similarly, we denote the long-tailed training set as $\hat{\mathcal{X}} = \{\hat{x}_i, \hat{y}_i\}, i \in \{1, \ldots, n\}$. Normally, we have $\forall i, p(\hat{y}_i) \neq p(y_i)$. Specifically, for a tail class $j$, $p(\hat{y}_j) \ll p(y_j)$, which makes the generalization under long-tailed scenarios extremely challenging.

We introduce Balanced Meta-Softmax (BALMS) for long-tailed visual recognition. It has two components: 1) a Balanced Softmax function that accommodates the label distribution shift between training and testing; 2) a Meta Sampler that learns to re-sample training set by meta-learning. We denote a feature extractor function as $f$ and a linear classifier's weight as $\theta$.

## 3.1 Balanced Softmax

**Label Distribution Shift.** We begin by revisiting the multi-class Softmax regression, where we are generally interested in estimating the conditional probability $p(y|x)$, which can be modeled as a multinomial distribution $\phi$:

$$\phi = \phi_1^{\mathbf{1}\{y=1\}} \phi_2^{\mathbf{1}\{y=2\}} \cdots \phi_k^{\mathbf{1}\{y=k\}}; \quad \phi_j = \frac{e^{\eta_j}}{\sum_{i=1}^{k} e^{\eta_i}}; \quad \sum_{j=1}^{k} \phi_j = 1 \tag{1}$$

where $\mathbf{1}(\cdot)$ is the indicator function and Softmax function maps a model's class-$j$ output $\eta_j = \theta_j^T f(x)$ to the conditional probability $\phi_j$.

From the Bayesian inference's perspective, $\phi_j$ can also be interpreted as:

$$\phi_j = p(y = j|x) = \frac{p(x|y = j)p(y = j)}{p(x)} \tag{2}$$

where $p(y = j)$ is in particular interest under the class-imbalanced setting. Assuming that all instances in the training dataset and the test dataset are generated from the same process $p(x|y = j)$, there could still be a discrepancy between training and testing given different label distribution $p(y = j)$ and evidence $p(x)$. With a slight abuse of the notation, we re-define $\phi$ to be the conditional distribution on the balanced test set and define $\hat{\phi}$ to be the conditional probability on the imbalanced training set. As a result, standard Softmax provides a biased estimation for $\phi$.

**Balanced Softmax.** To eliminate the discrepancy between the posterior distributions of training and testing, we introduce Balanced Softmax. We use the same model outputs $\eta$ to parameterize two conditional probabilities: $\phi$ for testing and $\hat{\phi}$ for training.

**Theorem 1.** *Assume $\phi$ to be the desired conditional probability of the balanced dataset, with the form $\phi_j = p(y = j|x) = \frac{p(x|y=j)}{p(x)} \frac{1}{k}$, and $\hat{\phi}$ to be the desired conditional probability of the imbalanced training set, with the form $\hat{\phi}_j = \hat{p}(y = j|x) = \frac{p(x|y=j)}{\hat{p}(x)} \frac{n_j}{\sum_{i=1}^{k} n_i}$. If $\phi$ is expressed by the standard Softmax function of model output $\eta$, then $\hat{\phi}$ can be expressed as*

$$\hat{\phi}_j = \frac{n_j e^{\eta_j}}{\sum_{i=1}^{k} n_i e^{\eta_i}}. \tag{3}$$

We use the exponential family parameterization to prove Theorem 1. The proof can be found in the supplementary materials. Theorem 1 essentially shows that applying the following Balanced Softmax function can naturally accommodate the label distribution shifts between the training and test sets. We define the Balanced Softmax function as

$$\hat{l}(\theta) = -\log(\hat{\phi}_y) = -\log\left(\frac{n_y e^{\eta_y}}{\sum_{i=1}^{k} n_i e^{\eta_i}}\right). \tag{4}$$

We further investigate the improvement brought by the Balanced Softmax in the following sections.

Many vision tasks, e.g., instance segmentation, might use multiple binary logistic regressions instead of a multi-class Softmax regression. By virtue of Bayes' theorem, a similar strategy can be applied to the multiple binary logistic regressions. The detailed derivation is left in the supplementary materials.

**Generalization Error Bound.** Generalization error bound gives the upper bound of a model's test error, given its training error. With dramatically fewer training samples, the tail classes have much higher generalization bounds than the head classes, which make good classification performance on tail classes unlikely. In this section, we show that optimizing Eqn. 4 is equivalent to minimizing the generalization upper bound.

Margin theory provides a bound based on the margins [15]. Margin bounds usually negatively correlate to the magnitude of the margin, i.e., a larger margin leads to lower generalization error. Consequently, given a constraint on the sum of margins of all classes, there would be a trade-off between minority classes and majority classes [4].

Locating such an optimal margin for multi-class classification is non-trivial. The bound investigated in [4] was established for binary classification using hinge loss. Here, we try to develop the margin bound for the multi-class Softmax regression. Given the previously defined $\phi$ and $\hat{\phi}$, we derive $\hat{l}(\theta)$ by minimizing the margin bound. Margin bound commonly bounds the 0-1 error:

$$err_{0,1} = \Pr\left[\theta_y^T f(x) < \max_{i \neq y} \theta_i^T f(x)\right]. \tag{5}$$

However, directly using the 0-1 error as the loss function is not ideal for optimization. Instead, negative log likelihood (NLL) is generally considered more suitable. With continuous relaxation of Eqn. 5, we have

$$err(t) = \Pr[t < \log(1 + \sum_{i \neq y} e^{\theta_i^T f(x) - \theta_y^T f(x)})] = \Pr\left[l_y(\theta) > t\right], \tag{6}$$

where $t \geq 0$ is any threshold, and $l_y(\theta)$ is the standard negative log-likelihood with Softmax, i.e., the cross-entropy loss. This new error is still a counter, but describes how likely the test loss will be larger than a given threshold. Naturally, we define our margin for class $j$ to be

$$\gamma_j = t - \max_{(x,y) \in S_j} l_j(\theta). \tag{7}$$

where $S_j$ is the set of all class $j$ samples. If we force a large margin $\gamma_j$ during training, i.e., force the training loss to be much lower than $t$, then $err(t)$ will be reduced. The Theorem 2 in [15] can then be directly generalized as

**Theorem 2.** *Let $t \geq 0$ be any threshold, for all $\gamma_j > 0$, with probability at least $1 - \delta$, we have*

$$err_{bal}(t) \lesssim \frac{1}{k} \sum_{j=1}^{k} \left(\frac{1}{\gamma_j}\sqrt{\frac{C}{n_j}} + \frac{\log n}{\sqrt{n_j}}\right); \quad \gamma_j^* = \frac{\beta n_j^{-1/4}}{\sum_{i=1}^{k} n_i^{-1/4}}, \tag{8}$$

*where $err_{bal}(t)$ is the error on the balanced test set, $\lesssim$ is used to hide constant terms and $C$ is some measure on complexity. With a constraint on $\sum_{j=1}^{k} \gamma_j = \beta$, Cauchy-Schwarz inequality gives us the optimal $\gamma_j^*$.*

The optimal $\gamma^*$ suggests that we need larger $\gamma$ for the classes with fewer samples. In other words, to achieve the optimal generalization ability, we need to focus on minimizing the training loss of the tail classes. To enforce the optimal margin, for each class $j$, the desired training loss $\hat{l}_j^*(\theta)$ is

$$\hat{l}_j^*(\theta) = l_j(\theta) + \gamma_j^*, \quad \text{where} \quad l_j(\theta) = -\log(\phi_j), \tag{9}$$

**Corollary 2.1.** *$\hat{l}_j^*(\theta) = l_j(\theta) + \gamma_j^* = l_j(\theta) + \frac{\beta n_j^{-1/4}}{\sum_{i=1}^{k} n_i^{-1/4}}$ can be approximated by $\hat{l}_j(\theta)$ when:*

$$\hat{l}_j(\theta) = -\log(\hat{\phi}_j); \quad \hat{\phi}_j = \frac{e^{\eta_j - \log \gamma_j^*}}{\sum_{i=1}^{k} e^{\eta_i - \log \gamma_i^*}} = \frac{n_j^{\frac{1}{4}} e^{\eta_j}}{\sum_{i=1}^{k} n_i^{\frac{1}{4}} e^{\eta_i}} \tag{10}$$

We provide a sketch of proof to the corollary in supplementary materials. Notice that compared to Eqn. 4, we have an additional constant $1/4$. We empirically find that setting $1/4$ to $1$ leads to the optimal results, which may suggest that Eqn. 8 is not necessarily tight. To this point, the label distribution shift and generalization bound of multi-class Softmax regression lead us to the same loss form: Eqn. 4.

## 3.2 Meta Sampler

**Re-sampling.** Although Balanced Softmax accommodates the label distribution shift, the optimization process is still challenging when given large datasets with extremely imbalanced data distribution. For example, in LVIS, the *bait* class may appear only once when the *banana* class appears thousands of times, making the *bait* class difficult to contribute to the model training due to low sample rate. Re-sampling is usually adopted to alleviate this issue, by increasing the number of minority class samples in each training batch. Recent works [29, 3] show that the global minimum of the Softmax regression is independent of the mini-batch sampling process. Our visualization in the supplementary material confirms this finding. As a result, a suitable re-sampling strategy could simplify the optimization landscape of Balanced Softmax under extremely imbalanced data distribution.

**Over-balance.** Class-balanced sampler (CBS) is a common re-sampling strategy. CBS balances the number of samples for each class in a mini-batch. It effectively helps to re-train the linear classifier in the decoupled training setup [16]. However, in our experiments, we find that naively combining CBS with Balanced Softmax may worsen the performance.

We first theoretically analyze the cause of the performance drop. When the linear classifier's weight $\theta_j$ for class $j$ converges, i.e., $\sum_{s=1}^{B} \frac{\partial L^{(s)}}{\partial \theta_j} = 0$, we should have:

$$\sum_{s=1}^{B} \frac{\partial L^{(s)}}{\partial \theta_j} = \sum_{s=1}^{B/k} f(x_{y=j}^{(s)})(1 - \hat{\phi}_j^{(s)}) - \sum_{i \neq j}^{k} \sum_{s=1}^{B/k} f(x_{y=i}^{(s)})\hat{\phi}_j^{(s)} = 0, \quad (11)$$

where $B$ is the batch size and $k$ is the number of classes. Samples per class have been ensured to be $B/k$ by CBS. We notice that $\hat{\phi}_j$, the output of Balanced Softmax, casts a varying, minority-favored effect to the importance of each class.

We use an extreme case to demonstrate the effect. When the classification loss converges to 0, the conditional probability of the correct class $\hat{\phi}_y$ is expected to be close to 1. For any positive sample $x^+$ and negative sample $x^-$ of class $j$, we have $\hat{\phi}_j(x^+) \approx \phi_j(x^+)$ and $\hat{\phi}_j(x^-) \approx \frac{n_j}{n_i}\phi_j(x^-)$, when $\hat{\phi}_y \to 1$. Eqn. 11 can be rewritten as

$$\frac{1}{n_j^2}\mathbb{E}_{(x^+,y=j) \sim D_{train}}[f(x^+)(1 - \phi_j)] - \sum_{i \neq j}^{k} \frac{1}{n_i^2}\mathbb{E}_{(x^-,y=i) \sim D_{train}}[f(x^-)\phi_j] \approx 0 \quad (12)$$

where $D_{train}$ is the training set. The formal derivation of Eqn. 12 is in the supplementary materials. Compared to the inverse loss weight, i.e., $1/n_j$ for class $j$, combining Balanced Softmax with CBS leads to the over-balance problem, i.e., $1/n_j^2$ for class $j$, which deviates from the optimal distribution.

Although re-sampling does not affect the global minimum, an over-balanced, tail class dominated optimization process may lead to local minimums that favor the minority classes. Moreover, Balanced Softmax's effect in the optimization process is dependent on the model's output, which makes hand-crafting a re-sampling strategy infeasible.

**Meta Sampler.** To cope with CBS's over-balance issue, we introduce Meta Sampler, a learnable version of CBS based on meta-learning, which explicitly learns the optimal sample rate. We first define the empirical loss by sampling from dataset $D$ as $L_D(\theta) = \mathbb{E}_{(x,y) \sim D}[l(\theta)]$ for standard Softmax, and $\hat{L}_D(\theta) = \mathbb{E}_{(x,y) \sim D}[\hat{l}(\theta)]$ for Balanced Softmax, where $\hat{l}(\theta)$ is defined previously in Eqn. 4.

To estimate the optimal sample rates for different classes, we adopt a bi-level meta-learning strategy: we update the parameter $\psi$ of sample distribution $\pi_\psi$ in the inner loop and update the classifier parameters $\theta$ in the outer loop,

$$\pi_\psi^* = \arg\min_\psi L_{D_{meta}}(\theta^*(\pi_\psi)) \quad s.t. \quad \theta^*(\pi_\psi) = \arg\min_\theta \hat{L}_{D_{q(x,y;\pi_\psi)}}(\theta), \quad (13)$$

where $\pi_\psi^j = p(y = j; \psi)$ is the sample rate for class $j$, $D_{q(x,y;\pi_\psi)}$ is the training set with class sample distribution $\pi_\psi$, and $D_{meta}$ is a meta set we introduce to supervise the inner loop optimization. We create the meta set by class-balanced sampling from the training set $D_{train}$. Empirically, we found it sufficient for inner loop optimization. An intuition to this bi-level optimization strategy is that: we

want to learn best sample distribution parameter $\psi$ such that the network, parameterized by $\theta$, outputs best performance on meta dataset $D_{meta}$ when trained by samples from $\pi_\psi$.

We first compute the per-instance sample rate $\rho_i = \pi_\psi^{c(i)}/n^{c(i)}$, where $c(i)$ denotes the class for instance $i$ and $n^{c(i)}$ is the number of samples in that class, and sample a training batch $B_\psi$ from a parameterized multi-nomial distribution $\rho$. Then we optimize the model in a meta-learning setup by

1. sample a mini-batch $B_\psi$ given distribution $\pi_\psi$ and perform one step gradient descent to get a surrogate model parameterized by $\tilde{\theta}$ by $\tilde{\theta} \leftarrow \theta - \nabla_\theta \hat{L}_{B_\psi}(\theta)$.

2. compute the $L_{D_{meta}}(\tilde{\theta})$ of the surrogate model on the meta dataset $D_{meta}$ and optimize the sample distribution parameter by $\psi \leftarrow \psi - \nabla_\psi L_{D_{meta}}(\tilde{\theta})$ with the standard cross-entropy loss with Softmax.

3. update the model parameter $\theta \leftarrow \theta - \nabla_\theta \hat{L}_{B_\psi}(\theta)$ with Balanced Softmax.

However, sampling from a discrete distribution is not differentiable by nature. To allow end-to-end training for the sampling process, when forming the mini-batch $B_\psi$, we apply the Gumbel-Softmax reparameterization trick [14]. A detailed explanation can be found in the supplementary materials.

## 4 Experiments

### 4.1 Exprimental Setup

**Datasets.** We perform experiments on long-tailed image classification datasets, including CIFAR-10-LT [18], CIFAR-100-LT [18], ImageNet-LT [23] and Places-LT [34] and one long-tailed instance segmentation dataset, LVIS [7]. We define the imbalance factor of a dataset as the number of training instances in the largest class divided by that of the smallest. Details of datasets are in Table 1.

**Evaluation Setup.** For classification tasks, after training on the long-tailed dataset, we evaluate the models on the corresponding balanced test/validation dataset and report top-1 accuracy. We also report accuracy on three splits of the set of classes: Many-shot (more than 100 images), Medium-shot ($20 \sim 100$ images), and Few-shot (less than 20 images). Notice that results on small datasets, i.e., CIFAR-LT 10/100, tend to show large variances, we report the mean and standard error under 3 repetitive experiments. We show details of long-tailed dataset generation in supplementary materials. For LVIS, we use official training and

| Dataset | #Classes | Imbalance Factor |
|---|---|---|
| CIFAR-10-LT [18] | 10 | 10-200 |
| CIFAR-100-LT [18] | 100 | 10-200 |
| ImageNet-LT [23] | 1,000 | 256 |
| Places-LT [34] | 365 | 996 |
| LVIS [7] | 1,230 | 26,148 |

Table 1: Details of long-tailed datatsets. For both CIFAR-10 and CIFAR-100, we report results with different imbalance factors.

testing splits. Average Precision (AP) in COCO style [21] for both bounding box and instance mask are reported. Our implementation details can be found in the supplementary materials.

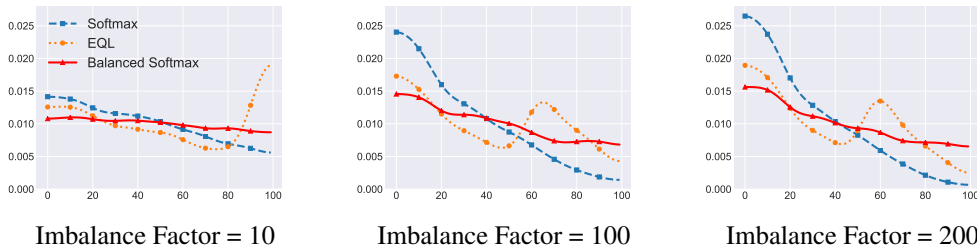

Figure 1: Experiment on CIFAR-100-LT. x-axis is the class labels with decreasing training samples and y-axis is the marginal likelihood $p(y)$ on the test set. We use end-to-end training for the experiment. Balanced Softmax is more stable under a high imbalance factor compared to the Softmax baseline and the SOTA method, Equalization Loss (EQL).

| Dataset | CIFAR-10-LT | | | CIFAR-100-LT | | |
|---|---|---|---|---|---|---|
| Imbalance Factor | 200 | 100 | 10 | 200 | 100 | 10 |
| End-to-end training | | | | | | |
| Softmax | $71.2 \pm 0.3$ | $77.4 \pm 0.8$ | $90.0 \pm 0.2$ | $41.0 \pm 0.3$ | $45.3 \pm 0.3$ | $61.9 \pm 0.1$ |
| CBW | $72.5 \pm 0.2$ | $78.6 \pm 0.6$ | $90.1 \pm 0.2$ | $36.7 \pm 0.2$ | $42.3 \pm 0.8$ | $61.4 \pm 0.3$ |
| CBS | $68.3 \pm 0.3$ | $77.8 \pm 2.2$ | $90.2 \pm 0.2$ | $37.8 \pm 0.1$ | $42.6 \pm 0.4$ | $61.2 \pm 0.3$ |
| Focal Loss [22] | $71.8 \pm 2.1$ | $77.1 \pm 0.2$ | $90.3 \pm 0.2$ | $40.2 \pm 0.5$ | $43.8 \pm 0.1$ | $60.0 \pm 0.6$ |
| Class Balanced Loss [6] | $72.6 \pm 1.8$ | $78.2 \pm 1.1$ | $89.9 \pm 0.3$ | $39.9 \pm 0.1$ | $44.6 \pm 0.4$ | $59.8 \pm 1.1$ |
| LDAM Loss [4] | $73.6 \pm 0.1$ | $78.9 \pm 0.9$ | $90.3 \pm 0.1$ | $41.3 \pm 0.4$ | $46.1 \pm 0.1$ | $62.1 \pm 0.3$ |
| Equalization Loss [30] | $74.6 \pm 0.1$ | $78.5 \pm 0.1$ | $90.2 \pm 0.2$ | $43.3 \pm 0.1$ | $47.4 \pm 0.2$ | $60.5 \pm 0.6$ |
| Decoupled training | | | | | | |
| cRT [16] | $76.6 \pm 0.2$ | $82.0 \pm 0.2$ | $91.0 \pm 0.0$ | $44.5 \pm 0.1$ | $50.0 \pm 0.2$ | $63.3 \pm 0.1$ |
| LWS [16] | $78.1 \pm 0.0$ | $83.7 \pm 0.0$ | $91.1 \pm 0.0$ | $45.3 \pm 0.1$ | $50.5 \pm 0.1$ | $\mathbf{63.4} \pm 0.1$ |
| BALMS | $\mathbf{81.5} \pm 0.0$ | $\mathbf{84.9} \pm 0.1$ | $\mathbf{91.3} \pm 0.1$ | $\mathbf{45.5} \pm 0.1$ | $\mathbf{50.8} \pm 0.0$ | $63.0 \pm 0.1$ |

Table 2: Top 1 accuracy for CIFAR-10/100-LT. Softmax: the standard cross-entropy loss with Softmax. CBW: class-balanced weighting. CBS: class-balanced sampling. LDAM Loss: LDAM loss without DRW. Results of Focal Loss, Class Balanced Loss, LDAM Loss and Equalization Loss are reproduced with optimal hyper-parameters reported in their original papers. BALMS generally outperforms SOTA methods, especially when the imbalance factor is high. Note that for all compared methods, we reproduce higher accuracy than reported in original papers. Comparison with their originally reported results is provided in the supplmentary materials.

## 4.2 Long-Tailed Image Classification

We present the results for long-tailed image classification in Table 2 and Table 3. On all datasets, BALMS achieves SOTA performance compared with all end-to-end training and decoupled training methods. In particular, we notice that BALMS demonstrates a clear advantage under two cases: 1) When the imbalance factor is high. For example, on CIFAR-10 with an imbalance factor of 200, BALMS is higher than the SOTA method, LWS [16], by 3.4%. 2) When the dataset is large. BALMS achieves comparable performance with cRT on ImageNet-LT, which is a relatively small dataset, but it significantly outperforms cRT on a larger dataset, Places-LT.

In addition, we study the robustness of the proposed Balanced Softmax compared to standard Softmax and SOTA loss function for long-tailed problems, EQL [30]. We visualize the marginal likelihood $p(y)$, i.e., the sum of scores on each class, on the test set with different losses given different imbalance factors in Fig. 1. Balanced Softmax clearly gives a more balanced likelihood under different imbalance factors. Moreover, we show Meta Sampler's effect on $p(y)$ in Fig. 2. Compared to CBS, Meta Sampler significantly relieves the over-balance issue.

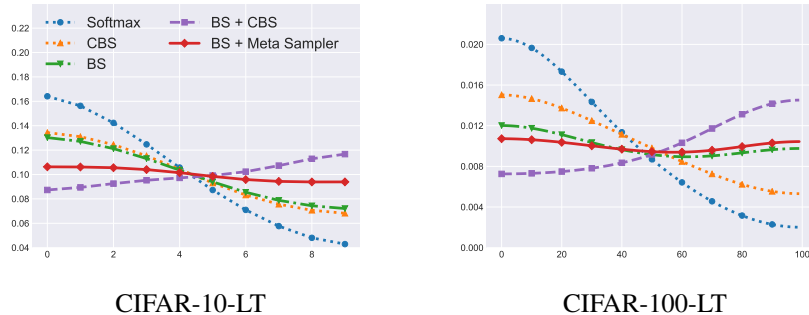

CIFAR-10-LT          CIFAR-100-LT

Figure 2: Visualization of $p(y)$ on test set with Meta Sampler and CBS. x-axis is the class labels with decreasing training samples and y-axis is the marginal likelihood $p(y)$ on the test set. The result is on CIFAR-10/100-LT with imbalance factor 200. We use decoupled training for the experiment. BS: Balanced Softmax. BS + CBS shows a clear bias towards the tail classes, especially on CIFAR-100-LT. Compared to BS + CBS, BS + Meta Sampler effectively alleviates the over-balance problem.

| Dataset | ImageNet-LT | | | | Places-LT | | | |
|---|---|---|---|---|---|---|---|---|
| Accuracy | Many | Medium | Few | Overall | Many | Medium | Few | Overall |
| **End-to-end training** | | | | | | | | |
| Lifted Loss [28] | 35.8 | 30.4 | 17.9 | 30.8 | 41.1 | 35.4 | 24.0 | 35.2 |
| Focal Loss [22] | 36.4 | 29.9 | 16.0 | 30.5 | 41.1 | 34.8 | 22.4 | 34.6 |
| Range Loss [33] | 35.8 | 30.3 | 17.6 | 30.7 | 41.1 | 35.4 | 23.2 | 35.1 |
| OLTR [23] | 43.2 | 35.1 | 18.5 | 35.6 | **44.7** | 37.0 | 25.3 | 35.9 |
| Equalization Loss [30] | - | - | - | 36.4 | - | - | - | - |
| **Decoupled training** | | | | | | | | |
| cRT [16] | - | - | - | **41.8** | 42.0 | 37.6 | 24.9 | 36.7 |
| LWS [16] | - | - | - | 41.4 | 40.6 | 39.1 | 28.6 | 37.6 |
| BALMS | **50.3** | **39.5** | **25.3** | **41.8** | 41.2 | **39.8** | **31.6** | **38.7** |

Table 3: Top 1 Accuracy on ImageNet-LT and Places-LT. We present results with ResNet-10 [23] for ImageNet-LT and ImageNet pre-trained ResNet-152 for Places-LT. Baseline results are taken from original papers. BALMS generally outperforms the SOTA models.

| Method | $AP_m$ | $AP_f$ | $AP_c$ | $AP_r$ | $AP_b$ |
|---|---|---|---|---|---|
| Softmax | 23.7 | 27.3 | 24.0 | 13.6 | 24.0 |
| Sigmoid | 23.6 | 27.3 | 24.0 | 12.7 | 24.0 |
| Focal Loss [22] | 23.4 | 27.5 | 23.5 | 12.8 | 23.8 |
| Class Balanced Loss [6] | 23.3 | 27.3 | 23.8 | 11.4 | 23.9 |
| LDAM [4] | 24.1 | 26.3 | 25.3 | 14.6 | 24.5 |
| LWS [16] | 23.8 | 26.8 | 24.4 | 14.4 | 24.1 |
| Equalization Loss [30] | 25.2 | 26.6 | 27.3 | 14.6 | 25.7 |
| Balanced Softmax[†] | 26.3 | **28.8** | 27.3 | 16.2 | 27.0 |
| BALMS | **27.0** | 27.5 | **28.9** | **19.6** | **27.6** |

Table 4: Results for LVIS dataset. $AP_m$ denotes Average Precision of masks. $AP_b$ denotes Average Precision of bounding box. $AP_f$, $AP_c$ and $AP_r$ denote Average Precision of masks on frequent classes, common classes and rare classes. †: the multiple binary logistic regression variant of Balanced Softmax, more details in the supplementary material. BALMS significantly outperforms SOTA models given high imbalance factor in LVIS. All compared methods are reproduced with higher AP than reported in the original papers.

## 4.3 Long-Tailed Instance Segmentation

LVIS dataset is one of the most challenging datasets in the vision community. As suggested in Tabel 1, the dataset has a much higher imbalance factor compared to the rest (26148 vs. less than 1000) and contains many very few-shot classes. Compared to the image classification datasets, which are relatively small and have lower imbalance factors, the LVIS dataset gives a more reliable evaluation of the performance of long-tailed learning methods.

Since one image might contain multiple instances from several categories, we hereby use Meta Reweighter, a re-weighting version of Meta Sampler, instead of Meta Sampler. As shown in Table 4, BALMS achieves the best results among all the approaches and outperform others by a large margin, especially in rare classes, where BALMS achieves an average precision of 19.6 while the best of the rest is 14.6. The results suggest that with the Balanced Softmax function and learnable Meta Reweighter, BALMS is able to give more balanced gradients and tackles the extremely imbalanced long-tailed tasks.

In particular, LVIS is composed of images of complex daily scenes with natural long-tailed categories. To this end, we believe BALMS is applicable to real-world long-tailed visual recognition challenges.

## 4.4 Component Analysis

We conduct an extensive component analysis on CIFAR-10/100-LT dataset to further understand the effect of each proposed component of BALMS. The results are presented in Table 5.

| Dataset | CIFAR-10-LT | | | CIFAR-100-LT | | |
|---|---|---|---|---|---|---|
| Imbalance Factor | 200 | 100 | 10 | 200 | 100 | 10 |
| End-to-end training | | | | | | |
| (1) Softmax | $71.2 \pm 0.3$ | $77.4 \pm 0.8$ | $90.0 \pm 0.2$ | $41.0 \pm 0.3$ | $45.3 \pm 0.3$ | $61.9 \pm 0.1$ |
| (2) Balanced Softmax $\frac{1}{4}$ | $71.6 \pm 0.7$ | $78.4 \pm 0.9$ | $90.5 \pm 0.1$ | $41.9 \pm 0.2$ | $46.4 \pm 0.7$ | $62.6 \pm 0.3$ |
| (3) Balanced Softmax | $79.0 \pm 0.8$ | $83.1 \pm 0.4$ | $90.9 \pm 0.4$ | $\mathbf{45.9} \pm 0.3$ | $50.3 \pm 0.3$ | $63.1 \pm 0.2$ |
| Decoupled training | | | | | | |
| (4) Balanced Softmax $\frac{1}{4}$+DT | $72.2 \pm 0.1$ | $79.1 \pm 0.2$ | $90.2 \pm 0.0$ | $42.3 \pm 0.0$ | $46.1 \pm 0.1$ | $62.5 \pm 0.1$ |
| (5) Balanced Softmax $\frac{1}{4}$+DT+MS | $76.2 \pm 0.4$ | $81.4 \pm 0.1$ | $91.0 \pm 0.1$ | $44.1 \pm 0.2$ | $49.2 \pm 0.1$ | $62.8 \pm 0.2$ |
| (6) Balanced Softmax+DT | $78.6 \pm 0.1$ | $83.7 \pm 0.1$ | $91.2 \pm 0.0$ | $45.1 \pm 0.0$ | $50.4 \pm 0.0$ | $63.4 \pm 0.0$ |
| (7) Balanced Softmax+CBS+DT | $80.6 \pm 0.1$ | $84.8 \pm 0.0$ | $91.2 \pm 0.1$ | $42.0 \pm 0.0$ | $47.4 \pm 0.2$ | $62.3 \pm 0.0$ |
| (8) DT+MS | $73.6 \pm 0.2$ | $79.9 \pm 0.4$ | $90.9 \pm 0.1$ | $44.2 \pm 0.1$ | $49.2 \pm 0.1$ | $63.0 \pm 0.0$ |
| (9) Balanced Softmax+DT+MR | $79.2 \pm 0.0$ | $84.1 \pm 0.0$ | $91.2 \pm 0.1$ | $45.3 \pm 0.3$ | $\mathbf{50.8} \pm 0.0$ | $\mathbf{63.5} \pm 0.1$ |
| (10) BALMS | $\mathbf{81.5} \pm 0.0$ | $\mathbf{84.9} \pm 0.1$ | $\mathbf{91.3} \pm 0.1$ | $45.5 \pm 0.1$ | $\mathbf{50.8} \pm 0.0$ | $63.0 \pm 0.1$ |

Table 5: Component Analysis on CIFAR-10/100-LT. CBS: class-balanced sampling. DT: decoupled training without CBS. MS: Meta Sampler. MR: Meta Reweighter. Balanced Softmax $\frac{1}{4}$: the loss variant in Eqn. 10. Balanced Softmax and Meta Sampler both contribute to the final performance.

**Balanced Softmax.** Comparing (1), (2) with (3), and (5), (8) with (10), we observe that Balanced Softmax gives a clear improvement to the overall performance, under both end-to-end training and decoupled training setup. It successfully accommodates the distribution shift between training and testing. In particular, we observe that Balanced Softmax $\frac{1}{4}$, which we derive in Eqn. 10, cannot yield ideal results, compared to our proposed Balanced Softmax in Eqn. 4.

**Meta-Sampler.** From (6), (7), (9) and (10), we observe that Meta-Sampler generally improves the performance, when compared with no Meta-Sampler, and variants of Meta-Sampler. We notice that the performance gain is larger with a higher imbalance factor, which is consistent with our observation in LVIS experiments. In (9) and (10), Meta-Sampler generally outperforms the Meta-Reweighter and suggests the discrete sampling process gives a more efficient optimization process. Comparing (7) and (10), we can see Meta-Sampler addresses the over-balancing issue discussed in Section 3.2.

**Decoupled Training.** Comparing (2) with (4) and (3) with (6), decoupled training scheme and Balanced Softmax are two orthogonal components and we can benefit from both at the same time.

# 5 Conclusion

We have introduced BALMS for long-tail visual recognition tasks. BALMS tackles the distribution shift between training and testing, combining meta-learning with generalization error bound theory: it optimizes a Balanced Softmax function which theoretically minimizes the generalization error bound; it improves the optimization in large long-tailed datasets by learning an effective Meta Sampler. BALMS generally outperforms SOTA methods on 4 image classification datasets and 1 instance segmentation dataset by a large margin, especially when the imbalance factor is high.

However, Meta Sampler is computationally expensive in practice and the optimization on large datasets is slow. In addition, the Balanced Softmax function only approximately guarantees a generalization error bound. Future work may extend the current framework to a wider range of tasks, e.g., machine translation, and correspondingly design tighter bounds and computationally efficient meta-learning algorithms.

# 6 Acknowledgements

This work is supported in part by the General Research Fund through the Research Grants Council of Hong Kong under grants (*Nos.* CUHK14208417, CUHK14207319), in part by the Hong Kong Innovation and Technology Support Program (*No.* ITS/312/18FX).

## Broader Impact

Due to the Zipfian distribution of categories in real life, algorithms, and models with exceptional performance on research benchmarks may not remain powerful in the real world. BALMS, as a light-weight method, only adds minimal computational cost during training and is compatible with most of the existing works for visual recognition. As a result, BALMS could be beneficial to bridge the gap between research benchmarks and industrial applications for visual recognition.

However, there can be some potential negative effects. As BALMS empowers deep classifiers with stronger recognition capability on long-tailed distribution, the application of such a classification algorithm can be further extended to more real-life scenarios. We should be cautious about the misuse of the method proposed. Depending on the scenario, it might cause negative effects on democratic privacy.

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
