[Supplementary Material]

# Supplementary Materials for
# Balanced Meta-Softmax
# for Long-Tailed Visual Recognition

**Jiawei Ren[1], Cunjun Yu[1], Shunan Sheng[1,2], Xiao Ma[1,3],**
**Haiyu Zhao[1*], Shuai Yi[1], Hongsheng Li[4]**
[1] SenseTime Research
[2] Nanyang Technological University
[3] National University of Singapore
[4] Multimedia Laboratory, The Chinese University of Hong Kong
{renjiawei, zhaohaiyu, yishuai}@sensetime.com    cunjun.yu@gmail.com
shen0152@e.ntu.edu.sg    xiao-ma@comp.nus.edu.sg    hsli@ee.cuhk.edu.hk

## A  Proofs and Derivations

### A.1  Proof to Theorem 1

The exponential family parameterization of the multinomial distribution gives us the standard Softmax function as the *canonical response function*

$$\phi_j = \frac{e^{\eta_j}}{\sum_{i=1}^{k} e^{\eta_i}} \tag{1}$$

and also the *canonical link function*

$$\eta_j = \log(\frac{\phi_j}{\phi_k}) \tag{2}$$

We begin by adding a term $-\log(\phi_j/\hat{\phi}_j)$ to both sides of Eqn. 2,

$$\eta_j - \log\frac{\phi_j}{\hat{\phi}_j} = \log(\frac{\phi_j}{\phi_k}) - \log(\frac{\phi_j}{\hat{\phi}_j}) = \log(\frac{\hat{\phi}_j}{\phi_k}) \tag{3}$$

Subsequently,

$$\phi_k e^{\eta_j - \log\frac{\phi_j}{\hat{\phi}_j}} = \hat{\phi}_j \tag{4}$$

$$\phi_k \sum_{i=1}^{k} e^{\eta_i - \log\frac{\phi_i}{\hat{\phi}_i}} = \sum_{i=1}^{k} \hat{\phi}_i = 1 \tag{5}$$

$$\phi_k = 1/\sum_{i=1}^{k} e^{\eta_i - \log\frac{\phi_i}{\hat{\phi}_i}} \tag{6}$$

Substitute Eqn. 6 back to Eqn. 4, we have

$$\hat{\phi}_j = \phi_k e^{\eta_j - \log\frac{\phi_j}{\hat{\phi}_j}} = \frac{e^{\eta_j - \log\frac{\phi_j}{\hat{\phi}_j}}}{\sum_{i=1}^{k} e^{\eta_i - \log\frac{\phi_i}{\hat{\phi}_i}}} \tag{7}$$

---

Recall that

$$\phi_j = p(y = j | x) = \frac{p(x | y = j)}{p(x)} \frac{1}{k}; \quad \hat{\phi}_j = \hat{p}(y = j | x) = \frac{p(x | y = j)}{\hat{p}(x)} \frac{n_j}{n} \tag{8}$$

then

$$\log \frac{\phi_j}{\hat{\phi}_j} = \log \frac{n}{k n_j} + \log \frac{\hat{p}(x)}{p(x)} \tag{9}$$

Finally, bring Eqn. 9 back to Eqn. 7

$$\hat{\phi}_j = \frac{e^{\eta_j - \log \frac{n}{k n_j} - \log \frac{\hat{p}(x)}{p(x)}}}{\sum_{i=1}^k e^{\eta_i - \log \frac{n}{k n_i} - \log \frac{\hat{p}(x)}{p(x)}}} = \frac{n_j e^{\eta_j}}{\sum_{i=1}^k n_i e^{\eta_i}} \tag{10}$$

## A.2 Derivation for the Multiple Binary Logisitic Regression variant

**Definition.** Multiple Binary Logisitic Regression uses $k$ binary logistic regression to do multi-class classification. Same as Softmax regression, the predicted label is the class with the maximum model output,

$$y_{pred} = \arg \max_j (\eta_j). \tag{11}$$

The only difference is that $\phi_j$ is expressed by a logistic function of $\eta_j$

$$\phi_j = \frac{e^{\eta_j}}{1 + e^{\eta_j}} \tag{12}$$

and the loss function sums up binary classification loss on all classes

$$l(\theta) = \sum_{j=1}^k - \log \tilde{\phi}_j \tag{13}$$

where

$$\tilde{\phi}_j = \begin{cases} \phi_j, & \text{if } y = j \\ 1 - \phi_j, & \text{otherwise} \end{cases} \tag{14}$$

**Setup.** By the virtue of Bayes' theorem, $\phi_j$ and $1 - \phi_j$ can be decomposed as

$$\phi_j = \frac{p(x | y = j) p(y = j)}{p(x)}; \quad 1 - \phi_j = \frac{p(x | y \neq j) p(y \neq j)}{p(x)} \tag{15}$$

and for $\hat{\phi}$ and $1 - \hat{\phi}$,

$$\hat{\phi}_j = \frac{p(x | y = j) \hat{p}(y = j)}{\hat{p}(x)}; \quad 1 - \hat{\phi}_j = \frac{p(x | y \neq j) \hat{p}(y \neq j)}{\hat{p}(x)} \tag{16}$$

**Derivation.** Again, we introduce the exponential family parameterization and have the following link function for $\phi_j$

$$\eta_j = \log \frac{\phi_j}{1 - \phi_j} \tag{17}$$

Bring the decomposition Eqn. 15 and Eqn.16 into the link function above

$$\eta_j = \log \left( \frac{\hat{\phi}_j}{1 - \hat{\phi}_j} \cdot \frac{\phi_j}{\hat{\phi}_j} \cdot \frac{1 - \hat{\phi}_j}{1 - \phi_j} \right) \tag{18}$$

$$\eta_j = \log \left( \frac{\hat{\phi}_j}{1 - \hat{\phi}_j} \cdot \frac{p(x | y = j) p(y = j) / p(x)}{p(x | y = j) \hat{p}(y = j) / \hat{p}(x)} \cdot \frac{p(x | y \neq j) \hat{p}(y \neq j) / \hat{p}(x)}{p(x | y \neq j) p(y \neq j) / p(x)} \right) \tag{19}$$

Simplify the above equation

$$\eta_j = \log \left( \frac{\hat{\phi}_j}{1 - \hat{\phi}_j} \cdot \frac{p(y = j)}{\hat{p}(y = j)} \cdot \frac{\hat{p}(y \neq j)}{p(y \neq j)} \right) \tag{20}$$

Substitute the $n_j$ in to the equation above

$$\eta_j = \log\left(\frac{\hat{\phi}_j}{1 - \hat{\phi}_j} \cdot \frac{n/k}{n_j} \cdot \frac{n - n_j}{n - n/k}\right) \tag{21}$$

Then

$$\eta_j - \log\left(\frac{n/k}{n_j} \cdot \frac{n - n_j}{n - n/k}\right) = \log\left(\frac{\hat{\phi}_j}{1 - \hat{\phi}_j}\right) \tag{22}$$

Finally, we have

$$\hat{\phi}_j = \frac{e^{\eta_j - \log\left(\frac{n/k}{n_j} \cdot \frac{n - n_j}{n - n/k}\right)}}{1 + e^{\eta_j - \log\left(\frac{n/k}{n_j} \cdot \frac{n - n_j}{n - n/k}\right)}} \tag{23}$$

**Remark.** A careful implementation should be made for instance segmentation tasks. As discussed in [17], suppressing background samples' gradient leads to a large number of false positives. Therefore, we restrict our loss to foreground samples, while applying the standard Sigmoid function to background samples, and ignore the constant $\frac{n/k}{n - n/k}$ to avoid penalizing the background class. Please refer to our code for the above-mentioned implementation details.

## A.3 Proof to Theorem 2

**Setup.** Firstly, we define $f$ as,

$$f(x) := -l(\theta) + t \tag{24}$$

where $l(\theta)$ and $t$ is previously defined in the main paper.

Let $err_j(t)$ be the 0-1 loss on example from class $j$

$$err_j(t) = \Pr_{(x,y) \in S_j}[f(x) < 0] = \Pr_{(x,y) \in S_j}[l(\theta) > t] \tag{25}$$

and $err_{\gamma,j}(t)$ be the 0-1 margin loss on example from class $j$

$$err_{\gamma,j}(t) = \Pr_{(x,y) \in S_j}[f(x) < \gamma_j] = \Pr_{(x,y) \in S_j}[l(\theta) + \gamma_j > t] \tag{26}$$

Let $\hat{err}_{\gamma,j}(t)$ denote the empirical variant of $err_{\gamma,j}(t)$.

**Proof.** For any $\delta > 0$ and with probability at least $1 - \delta$, for all $\gamma_j > 0$, and $f \in \mathcal{F}$, Theorem 2 in [7] directly gives us

$$err_j(t) \le \hat{err}_{\gamma,j}(t) + \frac{4}{\gamma_j}\hat{\mathfrak{R}}_j(\mathcal{F}) + \sqrt{\frac{\log(\log_2 \frac{4B}{\gamma_j})}{n_j}} + \sqrt{\frac{\log(1/\delta)}{2n_j}} \tag{27}$$

where $\sup_{(x,y) \in S}|l(\theta) - t| \le B$ and $\hat{\mathfrak{R}}_j(\mathcal{F})$ denotes the empirical Rademacher complexity of function family $\mathcal{F}$. By applying [1]'s analysis on the empirical Rademacher complexity and union bound over all classes, we have the generalization error bound for the loss on a balanced test set

$$err_{bal}(t) \le \frac{1}{k}\sum_{j=1}^{k}\left(\hat{err}_{\gamma,j}(t) + \frac{4}{\gamma_j}\sqrt{\frac{C(\mathcal{F})}{n_j}} + \epsilon_j(\gamma_j)\right) \tag{28}$$

where

$$\epsilon_j(\gamma_j) \triangleq \sqrt{\frac{\log(\log_2 \frac{4B}{\gamma_j})}{n_j}} + \sqrt{\frac{\log(1/\delta)}{2n_j}} \tag{29}$$

is a low-order term of $n_j$. To minimize the generalization error bound Eqn. 27, we essentially need to minimize

$$\sum_{j=1}^{k}\frac{4}{\gamma_j}\sqrt{\frac{C(\mathcal{F})}{n_j}} \tag{30}$$

By constraining the sum of $\gamma$ as $\sum_{j=1}^{k} \gamma_j = \beta$, we can directly apply Cauchy-Schwarz inequality to solve the optimal $\gamma$

$$\gamma_j^* = \frac{\beta n_j^{-1/4}}{\sum_{i=1}^{k} n_i^{-1/4}}. \tag{31}$$

## A.4 Proof to Corollary 2.1

**Preliminary.** Notice that $\hat{l}_j^*(\theta) = l_j(\theta) + \gamma_j^*$ can not be achieved for all class $j$, since $-\log \hat{\phi}_j^* = -\log \phi_j + \gamma_j^*$ and $\gamma_j^* > 0$ implies

$$\hat{\phi}_j^* < \phi_j; \quad \sum_{j=1}^{k} \hat{\phi}_j^* < \sum_{j=1}^{k} \phi_j = 1 \tag{32}$$

The equation above contradicts the definition that the sum of $\hat{\phi}^*$ should be exactly equal to 1. To solve the contradiction, we introduce a term $\gamma_{base} > 0$, such that

$$-\log \hat{\phi}_j^* = -\log \phi_j - \gamma_{base} + \gamma_j^*; \quad \sum_{j=1}^{k} \hat{\phi}_j^* = 1 \tag{33}$$

To justify the new term $\gamma_{base}$, we recall the definition of error

$$err_{\gamma,j}(t) = \Pr_{(x,y) \in S_j} [l(\theta) + \gamma_j > t]; \quad err_{bal}(t) = \Pr_{(x,y) \in S_{bal}} [l(\theta) > t] \tag{34}$$

If we tweak the threshold $t$ with the term $\gamma_{base}$

$$err_{\gamma,j}(t + \gamma_{base}) = \Pr_{(x,y) \in S_j} [l(\theta) + \gamma_j > t + \gamma_{base}] = \Pr_{(x,y) \in S_j} [(l(\theta) - \gamma_{base}) + \gamma_j > t] \tag{35}$$

$$err_{bal}(t + \gamma_{base}) = \Pr_{(x,y) \in S_{bal}} [l(\theta) > t + \gamma_{base}] = \Pr_{(x,y) \in S_{bal}} [(l(\theta) - \gamma_{base}) > t] \tag{36}$$

As $\gamma^*$ is not a function of $t$, the value of $\gamma^*$ will not be affected by the tweak. Thus, instead of looking for $\hat{l}_j^*(\theta) = l_j(\theta) + \gamma_j^*$ that minimizes the generalization bound for $err_{bal}(t)$, we are in fact looking for $\hat{l}_j^*(\theta) = (l_j(\theta) - \gamma_{base}) + \gamma_j^*$ that minimizes generalization bound for $err_{bal}(t + \gamma_{base})$

**Proof.** In this section, we show that $\hat{l}_j$ in the corollary is an approximation of $\hat{l}_j^*$.

$$\hat{l}_j(\theta) - (l_j(\theta) - \gamma_{base}) = \log \phi_j - \log \hat{\phi}_j + \gamma_{base} \tag{37}$$

$$= \log \frac{e^{\eta_j}}{\sum_{i=1}^{k} e^{\eta_i}} - \log \frac{e^{\eta_j - \log \gamma_j^*}}{\sum_{i=1}^{k} e^{\eta_i - \log \gamma_i^*}} + \gamma_{base} \tag{38}$$

$$= \log \frac{e^{\eta_j}}{\sum_{i=1}^{k} e^{\eta_i}} - \log \frac{e^{\eta_j}}{\sum_{i=1}^{k} e^{\eta_i - \log \gamma_i^* + \log \gamma_j^*}} + \gamma_{base} \tag{39}$$

$$= \log \sum_{i=1}^{k} e^{\eta_i - \log \gamma_i^* + \log \gamma_j^*} - \log \sum_{i=1}^{k} e^{\eta_i} + \gamma_{base} \tag{40}$$

$$= (\sum_{i=1}^{k} e^{\eta_i - \log \gamma_i^* + \log \gamma_j^*} - \sum_{i=1}^{k} e^{\eta_i})/\alpha + \gamma_{base} \quad \text{(Mean-Value Theorem)} \tag{41}$$

$$= (\gamma_j^* \sum_{i=1}^{k} \frac{1}{\gamma_i^*} e^{\eta_i} - \sum_{i=1}^{k} e^{\eta_i})/\alpha + \gamma_{base} \tag{42}$$

$$\geq (\frac{\gamma_j^*}{\beta} (\sum_{i=1}^{k} e^{\frac{1}{2}\eta_i})^2 - \sum_{i=1}^{k} e^{\eta_i})/\alpha + \gamma_{base} \quad \text{(Cauchy-Schwarz Inequality)} \tag{43}$$

$$= (\gamma_j^* \frac{\lambda}{\beta} \sum_{i=1}^{k} e^{\eta_i} - \sum_{i=1}^{k} e^{\eta_i})/\alpha + \gamma_{base} \quad (1 \leq \lambda \leq k) \tag{44}$$

$$\approx \gamma_j^* \quad (\text{let } \beta = 1, \gamma_{base} = 1) \tag{45}$$

$$\tag{46}$$

where $\alpha = \frac{d}{dx} \log(x')$ for some $x'$ in between $\sum_{i=1}^{k} e^{\eta_i - \log \gamma_i^* + \log \gamma_j^*}$ and $\sum_{i=1}^{k} e^{\eta_i}$, $\lambda$ is close to 1 when the model converges. Although the approximation holds under some constraints, we show that it approximately minimizes the generalization bound derived in the last section.

## A.5 Derivation for Eqn.12

Gradient for positive samples:

$$\frac{\partial \hat{l}_{y=j}^{(s)}(\theta)}{\partial \theta_j} = \frac{\partial - \log \hat{\phi}_j^{(s)}}{\partial \theta_j} \tag{47}$$

$$= \frac{\partial - \log \frac{e^{\theta_j^T f(x^{(s)}) + \log n_j}}{\sum_{i=1}^{n} e^{\theta_i^T f(x^{(s)}) + \log n_i}}}{\partial \theta_j} \tag{48}$$

$$= -\frac{\partial \theta_j^T f(x^{(s)}) + \log n_j}{\partial \theta_j} + \frac{\partial \log \sum_{i=1}^{n} e^{\theta_i^T f(x^{(s)}) + \log n_i}}{\partial \theta_j} \tag{49}$$

$$= -f(x^{(s)}) + f(x^{(s)}) \frac{e^{\theta_j^T f(x^{(s)}) + \log n_j}}{\sum_{i=1}^{n} e^{\theta_i^T f(x^{(s)}) + \log n_i}} \tag{50}$$

$$= -f(x^{(s)}) + f(x^{(s)}) \hat{\phi}_j^{(s)} \tag{51}$$

$$= f(x^{(s)})(\hat{\phi}_j^{(s)} - 1) \tag{52}$$

Gradient for negative samples:

$$\frac{\partial \hat{l}_{y\neq j}^{(s)}(\theta)}{\partial \theta_j} = \frac{\partial - \log \hat{\phi}_y^{(s)}}{\partial \theta_j} \tag{53}$$

$$= \frac{\partial - \log \frac{e^{\theta_y^T f(x^{(s)}) + \log n_y}}{\sum_{i=1}^{n} e^{\theta_i^T f(x^{(s)}) + \log n_i}}}{\partial \theta_j} \tag{54}$$

$$= -\frac{\partial \theta_y^T f(x^{(s)}) + \log n_y}{\partial \theta_j} + \frac{\partial \log \sum_{i=1}^{n} e^{\theta_i^T f(x^{(s)}) + \log n_i}}{\partial \theta_j} \tag{55}$$

$$= f(x^{(s)}) \frac{e^{\theta_j^T f(x^{(s)}) + \log n_j}}{\sum_{i=1}^{n} e^{\theta_i^T f(x^{(s)}) + \log n_i}} \tag{56}$$

$$= f(x^{(s)}) \hat{\phi}_j^{(s)} \tag{57}$$

Overall gradients on the training dataset:

$$\sum_{s=1}^{n} l^{(s)}(\theta) = \sum_{s=1}^{n_j} l_{y=j}^{(s)}(\theta) + \sum_{i\neq j}^{k} \sum_{s=1}^{n_i} l_{y=i}^{(s)}(\theta) \tag{58}$$

$$= \sum_{s=1}^{n_j} f(x^{(s)})(\hat{\phi}_j^{(s)} - 1) + \sum_{i\neq j}^{k} \sum_{s=1}^{n_i} f(x^{(s)}) \hat{\phi}_j^{(s)} \tag{59}$$

With Class-Balanced Sampling (CBS), number of samples in each class is equalized and therefore changed from $n_i$ and $n_j$ to $B/k$

$$\sum_{s=1}^{B} l^{(s)}(\theta) = \sum_{s=1}^{B/k} f(x^{(s)})(\hat{\phi}_j^{(s)} - 1) + \sum_{i\neq j}^{k} \sum_{s=1}^{B/k} f(x^{(s)}) \hat{\phi}_j^{(s)} \tag{60}$$

Set the overall gradient of a training batch to be zero gives

$$\sum_{s=1}^{B/k} f(x^{(s)})(1 - \hat{\phi}_j^{(s)}) - \sum_{i\neq j}^{k} \sum_{s=1}^{B/k} f(x^{(s)}) \hat{\phi}_j^{(s)} = 0 \tag{61}$$

We can also rewrite the equation using empirical expectation

$$\frac{1}{n_j} \mathbb{E}_{(x^+, y=j)\sim D_{train}}[f(x^+)(1 - \hat{\phi}_j)] - \sum_{i\neq j}^{k} \frac{1}{n_i} \mathbb{E}_{(x^-, y=i)\sim D_{train}}[f(x^-)\hat{\phi}_j] = 0 \tag{62}$$

Then we make the following approximation when the training loss is close to 0, i.e., $\hat{\phi}_y \to 1$

$$\lim_{\hat{\phi}_y \to 1} \frac{n_y e^{\eta_y}}{n_y e^{\eta_y} + \sum_{i\neq y}^{k} n_i e^{\eta_i}} = 1 \tag{63}$$

$$\lim_{\hat{\phi}_y \to 1} \frac{1}{1 + \sum_{i\neq y}^{k} \frac{n_i}{n_y} e^{\eta_i - \eta_y}} = 1 \tag{64}$$

$$\lim_{\hat{\phi}_y \to 1} \sum_{i\neq y}^{k} \frac{n_i}{n_y} e^{\eta_i - \eta_y} = 0 \tag{65}$$

$$\lim_{\hat{\phi}_y \to 1} \sum_{i\neq y}^{k} e^{\eta_i - \eta_y} = 0 \tag{66}$$

for positive samples:

$$\lim_{\hat{\phi}_{y=j}\to 1} \hat{\phi}_j/\phi_j = \lim_{\hat{\phi}_{y=j}\to 1} \frac{n_y e^{\eta_y}}{n_y e^{\eta_y} + \sum_{i\neq y}^k n_i e^{\eta_i}} / \frac{e^{\eta_y}}{e^{\eta_y} + \sum_{i\neq y}^k e^{\eta_i}} \tag{67}$$

$$= \lim_{\hat{\phi}_{y=j}\to 1} \frac{n_y e^{\eta_y}}{e^{\eta_y}} \cdot \frac{e^{\eta_y} + \sum_{i\neq y}^k e^{\eta_i}}{n_y e^{\eta_y} + \sum_{i\neq y}^k n_i e^{\eta_i}} \tag{68}$$

$$= \lim_{\hat{\phi}_{y=j}\to 1} n_y \cdot \frac{1}{n_y} \cdot \frac{1 + \sum_{i\neq y}^k e^{\eta_i-\eta_y}}{1 + \sum_{i\neq y}^k \frac{n_i}{n_y} e^{\eta_i-\eta_y}} \tag{69}$$

$$= \lim_{\hat{\phi}_{y=j}\to 1} n_y \cdot \frac{1}{n_y} \cdot \frac{1+0}{1+0} \tag{70}$$

$$= 1 \tag{71}$$

for negative samples:

$$\lim_{\hat{\phi}_{y\neq j}\to 1} \hat{\phi}_j/\phi_j = \lim_{\hat{\phi}_{y\neq j}\to 1} \frac{n_j e^{\eta_j}}{n_y e^{\eta_y} + \sum_{i\neq y}^k n_i e^{\eta_i}} / \frac{e^{\eta_j}}{e^{\eta_y} + \sum_{i\neq y}^k e^{\eta_i}} \tag{72}$$

$$= \lim_{\hat{\phi}_{y\neq j}\to 1} \frac{n_j e^{\eta_j}}{e^{\eta_j}} \cdot \frac{e^{\eta_y} + \sum_{i\neq y}^k e^{\eta_i}}{n_y e^{\eta_y} + \sum_{i\neq y}^k n_i e^{\eta_i}} \tag{73}$$

$$= \lim_{\hat{\phi}_{y\neq j}\to 1} n_j \cdot \frac{1}{n_y} \cdot \frac{1 + \sum_{i\neq y}^k e^{\eta_i-\eta_y}}{1 + \sum_{i\neq y}^k \frac{n_i}{n_y} e^{\eta_i-\eta_y}} \tag{74}$$

$$= \lim_{\hat{\phi}_{y\neq j}\to 1} n_j \cdot \frac{1}{n_y} \cdot \frac{1+0}{1+0} \tag{75}$$

$$= n_j/n_y \tag{76}$$

Therefore, when $\hat{\phi}_y \to 1$, Eqn.62 can be expanded as

$$\frac{1}{n_j}\mathbb{E}_{(x^+,y=j)\sim D_{train}}[f(x^+)(1-\phi_j)] - \sum_{i\neq j}^k \frac{1}{n_i}\mathbb{E}_{(x^-,y=i)\sim D_{train}}[f(x^-)\phi_j \frac{n_j}{n_i}] \approx 0 \tag{77}$$

That is

$$\frac{1}{n_j^2}\mathbb{E}_{(x^+,y=j)\sim D_{train}}[f(x^+)(1-\phi_j)] - \sum_{i\neq j}^k \frac{1}{n_i^2}\mathbb{E}_{(x^-,y=i)\sim D_{train}}[f(x^-)\phi_j] \approx 0 \tag{78}$$

# B   Detailed Description for Meta Sampler and Meta Reweighter

## B.1   Meta Sampler

To estimate the optimal sample rate, we first make the sampler differentiable. Normally, class-balanced samplers take following steps:

1. Define a class sample distribution $\pi = \pi_1^{\mathbf{1}\{y=1\}}\pi_2^{\mathbf{1}\{y=2\}}\ldots\pi_k^{\mathbf{1}\{y=k\}}$.
2. Assign $\pi_j$ to all instance-label pairs $(x, y = j)$ and normalize over the dataset, to give the instance sample distribution $\rho = \rho_1^{\mathbf{1}\{i=1\}}\rho_2^{\mathbf{1}\{i=2\}}\ldots\rho_n^{\mathbf{1}\{i=n\}}$.
3. Draw discrete image indexes from $\rho$ to form a batch with a size $b$.
4. Augment the images and feed images into a model.

The steps where discrete sampling and image augmentation happen are usually not differentiable. We propose a simple yet effective method to back-propagate the gradient directly from the loss to the learnable sample rates.

Firstly, we use the Straight-through Gumbel Estimator [6] to approximate the gradient through the multinomial sampling:

$$s_j = \frac{((\log \rho_j + g_j)/\tau)}{\sum_{i=1}^n \exp((\log(\rho_i + g_i)/\tau))} \tag{79}$$

where $s$ is the sample result, $g$ is i.i.d. samples drawn from Gumbel$(0, 1)$ and $\tau$ is the temperature coefficient. Straight-through means that we use argmax to discretize $s$ to (0,1) during forward and use $\nabla s$ during backward. Gumbel-Softmax re-parameterization is commonly found to have less variance in gradient estimation than score functions [6].

Then, we use an external memory to connect sampler with loss. We use the Straight-through Gumbel Estimator to draw $b$ discrete samples from $\rho$, we denote as $s^{b \times n}$. $s^{b \times n}$ is matrix of a $n$-dimensional one-hot vectors, representing $b$ selected images. Concretely, for the $i$-th sample, if the Gumbel Estimator gives a sampling result to be $c$-th image, we have $s^{(i)}$ to be

$$s_j^{(i)} = \begin{cases} 1, & \text{if } j = c \\ 0, & \text{otherwise} \end{cases} \tag{80}$$

We save this matrix into an external memory during data preparation. After obtaining the classification loss $l(\theta)$, which is the $i$-th loss in the batch computed from the $c$-th sample, we re-weight the loss by

$$\tilde{l}^{(i)}(\theta) = l^{(i)}(\theta) \cdot s_c^{(i)} \tag{81}$$

Notice that the re-weight will not change the loss value, it only connects sampling results with the classification loss in the computation graph. By doing so, the gradient from the loss can directly reach the learnable sample rate $\pi$.

### B.2 Meta Reweighter

Since one image might contain multiple instances from several categories, we use Meta Reweighter, rather than Meta Sampler on the LVIS dataset. Specifically, we assign the loss weight for instance $i$ to be $\rho_i = \pi_j$, where $\pi$ is a learnable class weight and $j$ is the class label of instance $i$. Next, we perform similar bi-level optimization as in Meta Sampler, where we re-weight the loss of an instance by its loss weight $\rho_i$ instead of a discrete 0-1 sampling result $s_i$.

## C  Implementation Details

### C.1  Hardware

We use Intel Xeon Gold 6148 CPU @ 2.40GHz with Nvidia V100 GPU for model training. We take a single GPU to train models on CIFAR-10-LT, CIRFAR-100-LT, ImageNet-LT and Places-LT, and 8 GPUs to train models on LVIS.

### C.2  Software

We implement our proposed algorithm with PyTorch-1.3.0 [14] for all experiments. Second-order derivatives are computed with Higher [4] library.

### C.3  Training details

**Decoupled Training.** Through the paper, we refer to decoupled training as training the last linear classifier on a fixed feature extractor obtained from instance-balanced training.

**Meta Sampler/Reweighter.** We apply Meta Sampler/Reweighter only when decoupled training to save computational costs. We start them at the beginning of the decoupled training with no deferment.

**CIFAR-10-LT and CIFAR-100-LT.** All experiments use ResNet-32 as backbone like [2]. We use Nesterov SGD with momentum 0.9 and weight-decay 0.0005 for training. We use a total mini-batch size of 512 images on a single GPU. The learning rate increased from 0.05 to 0.1 in the first 800 iterations. Cosine scheduler [13] is applied afterward, with a minimum learning rate of 0. Our augmentation follows [17]. In testing, the image size is 32x32. In end-to-end training, the model

is trained for 13K iterations. In decoupled training experiments, we fix the Softmax model, i.e., the instance-balanced baseline model obtained from the previous end-to-end training, as the feature extractor. And the classifier is trained for 2K iterations. For Meta Sampler and Meta Reweighter, we use Adam[9] with betas (0.9, 0.99) and weight decay 0. The learning rate is set to 0.01 with no warm-up strategy or scheduler applied. The meta-set is formed by randomly sampling 512 images from the training set with replacement, using Class-Balanced Sampling.

**ImageNet-LT and Places-LT.** We follow the setup in [8] for decoupled classifier retraining. We first train a base model without any bells and whistles following Kang et al. [8] for these two datasets. For ImageNet-LT, the model is trained for 90 epochs from scratch. For Places-LT, we choose ResNet-152 as the backbone network pre-trained on the full ImageNet-2012 dataset and train it on Places-LT following Kang et al [8]. For both datasets, we use SGD optimizer with momentum 0.9, batch size 512, cosine learning rate schedule [13] decaying from 0.2 to 0 and image resolution $224 \times 224$.

After obtaining the base model, we retrain the last linear classifier. For Meta Sampler, we use Adam[9] with betas (0.9, 0.99) and weight decay 0. The learning rate is set to 0.01 with no warm-up strategy and is kept unchanged during the training process. The meta-set is formed by randomly sampling 512 images from the training set with replacement, using Class-Balanced Sampling. For ImageNet-LT, we use SGD optimizer with momentum 0.9, batch size 512, cosine learning rate schedule decaying from 0.2 to 0 for 10 epochs. For Places-LT, we use SGD optimizer with momentum 0.9, batch size 128, cosine learning rate schedule decaying from 0.01 to 0 for 10 epochs.

For the training process, we resize the image to $224 \times 224$. During testing, we first resize the image to $256 \times 256$ and do center-crop to obtain an image of $224 \times 224$.

**LVIS.** We use the off-the-shelf model Mask R-CNN with the backbone network ResNet-50 for LVIS. The backbone network is pre-trained on ImageNet. We follow the setup (including Repeat Factor Sampling) from the original dataset paper [5] for two baseline models (Softmax and Sigmoid). We use an SGD optimizer with 0.9 momentum, 0.01 initial learning rate, and 0.0001 weight decay. The model is trained for 90k iterations with 8 images per mini-batch. The learning rate is dropped by a factor of 10 at both 60k iterations and 80k iterations.

Methods other than baselines are trained under the decoupled training scheme, with the above-mentioned models as the base model. Slightly different from the decoupled training for classification tasks [8], we fine-tune the bounding box classifier (one fully connected layer) instead of retraining it from scratch. This significantly saves the training time. We use an SGD optimizer with 0.9 momentum, 0.02 initial learning rate, and 0.0001 weight decay. The model is trained for 22k iterations with 8 images per mini-batch. The learning rate is dropped by a factor of 10 at both 11k iterations and 18k iterations.

For our method with a Meta Reweighter, we use Adam optimizer with 0.001 for the Meta Reweighter and train the Meta Reweighter together with the model. The learning rate is kept unchanged during the training process.

We apply scale jitter and random flip at training time (sampling image scale for the shorter side from 640, 672, 704, 736, 768, 800). For testing, images are resized to a shorter image edge of 800 pixels; no test-time augmentation is used.

### C.4 Meta-learned sample rates with Softmax and Balanced Softmax

Figure 1 demonstrates that compared with standard Softmax function, Meta Sampler learns a more *balanced* sample rates with our proposed Balanced Softmax. The sample rates for all the classes are initialized with 0.5 and are constrained in the range of (0,1).

The blue bar represents the learned sample rates with standard Softmax. The sample rates of tail classes approach 1 while the sample rates of head classes approach 0. Such an extreme divergence in sample rates could potentially pose challenges to the meta-learning optimization process. A very low optimal learning rate may also not be numerically stable.

With Balanced Softmax, we can see that Meta Sampler produces a more balanced distribution of sample rates. After convergence, the sample rates for Softmax has a variance of 0.13. Balanced Softmax significantly reduces the variance to 0.03.

Figure 1: Learned sample rates with Meta-Sampler when training with Softmax and Balanced Softmax. The experiment is on CIFAR-100-LT with imbalanced factor 200. The X-axis denotes classes with a decreasing number of training samples. Y-axis denotes sample rates for different classes. Balanced Softmax gives a smoother distribution compared to Softmax.

## D    More Details Regarding Datasets

### D.1    Basic information

We hereby provide more details about datasets mentioned in the paper in Table 1

| Dataset | #Classes | Imbalance Factor | #Train Instances | Head Class Size | Tail Class Size |
|---------|----------|------------------|------------------|-----------------|-----------------|
| CIFAR-10-LT [10] | 10 | 10-200 | 50,000 – 11,203 | 5,000 | 500-25 |
| CIFAR-100-LT [10] | 100 | 10-200 | 50,000 – 9,502 | 500 | 50-2 |
| ImageNet-LT [12] | 1,000 | 256 | 115,846 | 1280 | 5 |
| Places-LT [18] | 365 | 996 | 62,500 | 4,980 | 5 |
| LVIS [5] | 1,230 | 26,148 | 693,958 | 26,148 | 1 |

Table 1: Details of long-tailed datatsets. Notice that for both CIFAR-10-LT and CIFAR-100-LT, the number of tail class varies with different imbalance factors.

All the datasets are publicly available for downloading, we provide the download link as follows: ImageNet, CIFAR-10 and CIFAR-100, Places365, and LVIS.

### D.2    Long-tailed datasets generation

**CIFAR10-LT and CIFAR100-LT.** We generated the long-tailed version of CIFAR-10 and CIFAR-100 following Cui et al. [2]. For both the original CIFAR-10 and CIFAR-100, they contain 50000 training images and 10000 test images at a size of $32 \times 32$ uniformly distributed in 10 classes and 100 classes. The long-tailed version is created by randomly reducing training samples. In particular, the number of samples in the y-th class is $n_y \mu^y$, where $n_y$ is the original number of training samples in the class and $\mu \in (0, 1)$. By varying $\mu$, we generate three training sets with the imbalance factors of 200, 100, and 10. The test set is kept unchanged and balance.

**ImageNet-LT.** We use the long-tailed version of ImageNet from Liu et al. [12]. It is created by firstly sampling the class sizes from a Pareto distribution with the power value $\alpha = 6$, followed by sampling the corresponding number of images for each class. The ImageNet-LT dataset has 115,846

| Dataset | CIFAR-10-LT | | | CIFAR-100-LT | | |
|---|---|---|---|---|---|---|
| Imbalance Factor | 200 | 100 | 10 | 200 | 100 | 10 |
| Focal Loss* [11] | 65.29 | 70.38 | 86.66 | 35.62 | 38.41 | 55.78 |
| Class Balanced Loss* [2] | 68.89 | 74.57 | 87.49 | 36.23 | 39.60 | 57.99 |
| L2RW* [15] | 66.51 | 74.16 | 85.19 | 33.38 | 40.23 | 53.73 |
| LDAM† [1] | - | 73.35 | 86.96 | - | 39.6 | 56.91 |
| LDAM-DRW† [1] | - | 77.03 | 88.16 | - | 42.04 | 58.71 |
| Meta-Weight-Net* [16] | 68.89 | 75.21 | 87.84 | 37.91 | 42.09 | 58.46 |
| Equalization Loss‡ [17] | - | - | - | 43.38 | - | - |
| BALMS | **81.5** | **84.9** | **91.3** | **45.5** | **50.8** | **63.0** |

Table 2: Comparisons with reported SOTA results on Top 1 accuracy for CIFAR-LT. * indicates results reported in [16]. † indicates results reported in [1]. ‡ indicates results reported in [17].

| Feature Training | Classifier Training | Accuracy |
|---|---|---|
| Softmax | Softmax | **69.53** |
| Softmax+CBS | Softmax | 57.06 |
| Balanced Softmax | Softmax | 65.75 |
| Softmax | Softmax+CBS | **76.59** |
| Softmax+CBS | Softmax+CBS | 63.96 |
| Balanced Softmax | Softmax+CBS | 75.35 |
| Softmax | Balanced Softmax | **78.53** |
| Softmax+CBS | Balanced Softmax | 68.24 |
| Balanced Softmax | Balanced Softmax | 77.04 |

Table 3: Comparison of decoupled training results with features from Softmax and Balanced Softmax. The experiment is on CIFAR-10-LT with imbalanced factor 200. The Softmax pretrained features generally outperform the Balanced Softmax pretrained features.

training images in 1,000 classes, and its imbalance factor is 256 as shown in Table 1. The original ImageNet [3] validation set is used as the test set, which contains 50 images for each class.

**Places-LT.** In a similar spirit to the long-tailed ImageNet, a long-tailed version of the Places-365 dataset is generated using the same strategy as above. It contains 62,500 training images from 365 classes with an imbalance factor 996. In the test set, there are 100 test images for each class.

**LVIS.** We use official training and validation split from LVIS [5]. No modification is made.

# E   Comparisons with Reported SOTA Results on CIFAR-LT

We used our reproduced results on CIFAR-LT in the empirical analysis section in the paper since prior works chose different baselines and cannot be fairly compared with. Table 2 compares our method with more results originally reported in corresponding papers.

# F   More Visualizations and Analysis

## F.1   Visualization and analysis on the feature space of Balanced Softmax

Recent work [8] shows that instance-balanced training results in the best feature space in practice. In this section, we use t-SNE to visualize the feature space created by Balanced Softmax. The result is shown in Fig. 2. The following pattern can be observed: CBS and Balanced Softmax tend to have a more concentrated center area compared to the Softmax baseline. This indicates that the Softmax baseline's features are more suitable for the classification task than Balanced Softmax and CBS's. Further empirical analysis in Table 3 advocates the claim.

Softmax          Softmax+CBS          Balanced Softmax

Figure 2: t-SNE visualization of the feature space created by different methods. The experiment is on CIFAR-10-LT with imbalanced factor 200. The 10 colors represent the 10 classes. Compared to Softmax, Softmax+CBS and Balanced Softmax have a more concentrated center area, making them less suitable for classification.

## F.2   Visualization of re-sampling's effect towards training

We use a two-dimensional, three-way classification example to demonstrate re-sampling's effect on training a one-layer linear classifier either with standard Softamx or with Balanced Softmax. The result, shown in Figure 3, confirms that the linear classifier's solution is unaffected by re-sampling. Meanwhile, different re-sampling strategies have different effects on the optimization process, where CBS causes the over-balance problem to Balanced Softmax's optimization.

Figure 3: Visualization of decision boundaries over iterations with different training setups. We create an imbalanced, two-dimensional, dummy dataset of three classes: red, yellow and blue. The red point represents 10000 red samples, the yellow point represents 100 yellow samples and the blue point represents 1 blue sample. Background shading shows the decision surface. Both Softmax and Softmax+CBS converge to symmetric decision boundaries, and Softmax+CBS converges faster than Softmax. Note that symmetric decision boundaries do not optimize for the generalization error bound on an imbalanced dataset [1]. Both Balanced Softmax and Balanced Softmax+CBS converge to a better solution: they successfully push the decision boundary from the minority class toward the majority class. Compared to Balanced Softmax, Balanced Softmax+CBS shows the over-balance problem: its optimization is dominated by the minority class.