[Reviews · NeurIPS 2020]

Review 1

Summary and Contributions: The authors first showed that Softmax function gives a biased gradient estimation under the long-tailed setup, and presented Balanced Softmax to accommodate the label distribution shift between training and testing. Theoretically, the authors derived the generalization bound for multiclass Softmax regression and show our loss minimizes the bound. Also, they introduced Balanced Meta-Softmax, applying a complementary Meta Sampler to estimate the optimal class sample rate and further improve long-tailed learning. In the experiments, they demonstrate that Balanced Meta-Softmax outperforms state-of-the-art long-tailed classification solutions on both visual recognition and instance segmentation tasks.

Strengths: - Balanced Softmax, a simple extension of softmax, is easy to implement yet effective. - The authors provided theoretical proof of the advantages of the proposed method. - The authors provided an ablation study to demonstrate the effectiveness of each component and achieved state-of-the-art performance in various datasets.

Weaknesses: - Empirically, “Balanced Softmax 1/4” (Eqn(10)) couldn’t achieve better performance than “Balanced Softmax” (Eqn(4)) which conflicts with the message from Eqn (8). As Theorem 2 and Corollary 2.1 are the important contributions in this paper, there must be a convincing explanation about the mismatch between the theory and practice. - It would be better to evaluate the proposed model on other datasets with long-tailed distribution such as iNaturalist or Open Images Dataset.

Correctness: It seems the claims, method and empirical methodology are correct.

Clarity: This paper is well written and the proposed method is easy to understand.

Relation to Prior Work: It seems the paper clearly referred and discussed the difference from previous methods.

Reproducibility: Yes

Additional Feedback: - Suggestion: It would be an interesting future direction to show a mathematical link between the balanced softmax and the other traditional approaches such as loss re-weighting or balanced sampling. ######################### Comments after rebuttal After reading the author's responses and the other reviewers' comments, I decided to keep my decision for accept.


Review 2

Summary and Contributions: The paper studies imbalanced/long-tailed classification. The authors found that the widely-used softmax function gives a biased gradient estimator under the imbalanced/long-tailed setup. They proposed a simple modification by adding a class-dependent constant term to each class (according to the class size) in the softmax function. They further develop a meta sampler for further improvement. ---------------------------- Post rebuttal comments ------------------------- Please see 8. additional feedback.

Strengths: -The proposed "balanced softmax" in (3) and (4) is well-motivated, described, and mathematically founded. The method can easily be implemented/reproduced and be combined with different network architecture (good applicability). -The authors further provide a theoretical analysis of the proposed approach and the generalization bound. -The derivation in section 3.2 (Line 151-165) is inspiring, although the conclusion is somewhat expected. As balanced softmax already "balances" the imbalanced training, further "balancing" the training by re-sampling will over-balance the training. -The authors performed extensive empirical studies on a variety of datasets and tasks, including object classification and instance segmentation. The authors further pointed out the weakness to improve (in Conclusion), which is highly encouraged and appreciated.

Weaknesses: -The equations (3) and (4) are, however, very similar to [3] and [A, B] in the way that they force the minor-class examples to have larger decision values (i.e., \exp \eta_j) in training. The proposed softmax seems particularly similar to eq. (11) in [B]. The authors should have cited these papers and provided further discussion and comparison. This point limits the novelty/significance of the paper. [A] Han-Jia Ye et al., Identifying and Compensating for Feature Deviation in Imbalanced Deep Learning, arxiv 2020 [B] S.H. Khan et al., Cost-Sensitive Learning of Deep Feature Representations from Imbalanced Data, IEEE transactions on neural networks and learning systems, 2017. -The proposed meta sampler has a similar idea to [12,24,27] but the authors didn't differentiate the proposed method from theirs. It is hard for me to judge the novelty of the proposed meta sampler. (It is indeed quite similar to [24].) These methods are also not compared in the experiments. From Table 5, I don't quite see the benefit of the meta sampler over the meta reweighter. -The loss function derived from the theoretical analysis seems to not directly imply the proposed softmax. In eq. (9), the "margin" term \gamma^\star_j is a class-dependent constant, and adding them into the overall loss won't affect the learning of the network parameters. Nevertheless, in equation (10), such a "margin" term becomes affective to the network training. -I don't quite follow why the authors want to bring in class-balanced sampler or meta-sampler. The authors argued that re-sampling techniques can be harmful to model training (see Line 23-26, 50-56), but finally still apply it. I would suggest that the authors provide more discussions about why it is needed in extremely imbalanced cases. Moreover, the description of the meta sampler is a bit hard to follow: 1) Is the sample distribution updated in the inner loop or outer? From Line 171-175, it seems that the outer loop will update the sample distribution. 2) Do the authors only apply the meta sampler in a decoupled way? That is, to update the linear classifier when the features are fixes? If so, please provide more discussion on this and when (which epoch) do the authors start applying the meta sampler? 3) The addition of the meta sampler makes the contributions of the paper a bit vague: please include both the balanced softmax results w/ and w/o the meta sampler in the experimental results. If the meta sampler is used in a decoupled way, then when to start the meta sampler leads to a mother hyper-parameter. Also, the authors mentioned that on LVIS, they used meta reweighter rather than meta sampler, which is confusing. -Last but not the least, in the experiments the authors' baseline softmax results (in Table 2) are much higher than those reported in other papers. The baseline results are even better than or on part with existing approaches. I thus doubt if the superb performance reported by the authors is partly due to a better baseline method.

Correctness: The proposed methods are mostly correct. I list some questions as follows. -Does (8) imply that the training error/loss is 0, as there is no term related to the empirical error on the right-handed side? If so, can this be always achieved in imbalanced/long-tailed classification? If not, how will the generalization bound be modified to incorporate the training error/loss? -The theoretical analysis does not directly align with the proposed softmax. See above for more details.

Clarity: -The paper is mostly well-written, while section 3 and the detail of the proposed approach can be improved. -Line 115-136 can be improved. Please clearly indicate which terms are for the error on the test data distribution, and which are for the empirical error estimated on the imbalanced training data. What is the definition of "l_j" and what does the desired training error "l^\star_j" mean? -Line 145-149 is a bit confusing. Do the authors finally apply eq. (4) or eq. (10)? "Notice that compared to Eqn. 4 ... suggest that Eqn. 8 is not necessarily tight." is confusing. The authors said setting 1/4 (which is indicated by eq. (10) and (8)) leads to the optimal results, but also said that eq. (8) is not necessarily tight. -How is the marginal likelihood p(y) at Line 228 computed? -How is the meta sampler applied? In a decoupled way or not?

Relation to Prior Work: -The following discussions are missing. -The proposed softmax in equations (3) and (4) are very similar to [3] and [A, B] in the way that they force the minor-class examples to have larger decision values (i.e., \exp \eta_j) in training. The proposed softmax seems particularly similar to eq. (11) in [B]. [A] Han-Jia Ye et al., Identifying and Compensating for Feature Deviation in Imbalanced Deep Learning, arxiv 2020 [B] S.H. Khan et al., Cost-Sensitive Learning of Deep Feature Representations from Imbalanced Data, IEEE transactions on neural networks and learning systems, 2017. -The proposed meta sampler has a similar idea to [12,24,27] but the authors didn't differentiate the proposed method from theirs. It is hard for me to judge the novelty of the proposed meta sampler. (It is indeed quite similar to [24].) These methods are also not compared in the experiments. From Table 5, I don't quite see the benefit of the meta sampler over the meta reweighter.

Reproducibility: No

Additional Feedback: -If the class proportion between the training/testing is the main problem (section 3.1), can re-sampling/re-weighting solve this problem (i.e., making the training prior to be equal over all classes), why and why not? -What is exactly the DT in Table 5? Just re-train the last layer without balanced sampling? -It will be great if the authors can also discuss this paper, which focuses on generalization for minor and major groups. Shiori Sagawa et al., Distributionally robust neural networks for group shifts: On the importance of regularization for worst-case generalization, ICLR 2020. -I would suggest that the authors improve section 3 and section 4. Please clearly mention when the meta sampler is used and what the benefit of it over a meta reweighter is. Please include the results by the balanced softmax w/ and w/o the meta sampler to clearly show the benefit of the balanced softmax. -I would suggest that the authors also work on the iNaturalist dataset. -Can the authors discuss if applying post-calibration like [15], [A], [C], [D] can also re-balance the biased softmax? [C] Mateusz Buda et al., A systematic study of the class imbalance problem in convolutional neural networks. Neural Networks, 2018 [D] B. Kim and J. Kim, Adjusting decision boundary for class imbalanced learning, IEEE Access 2020. --------------------- Post rebuttal comments ---------------------------- I read the authors' responses and I appreciate their efforts, especially on providing further explanations and comparisons. Nevertheless, I don't think my concerns are fully addressed. -Novelty on Meta-sampling: As in Table 5, Meta-sampling only notably improves Meta-reweighting on CIFAR-10-200 (just 1 out of 6 cases). The authors also applied Meta-reweighting to LVIS, but not Meta-sampling (Line 236-237), which is confusing. Thus, I don't think Meta-sampling outperforms Meta-reweighting. The similarity to [12,24,27], especially [12] which also works on imbalanced learning, further limits the novelty of the proposed Meta-sampling. The authors should compare their method to [12] in their experiments. -Not convincing comparisons: The facts that the authors used a stronger softmax baseline and that many existing methods show no improvement against that baseline make Table 2 a bit not convincing. Specifically, LDAM [3] has nearly no improvement, even after the authors retrained LDAM using the stronger baseline. I'm not sure if the authors tuned the hyper-parameter of LDAM or if the authors applied the stronger version LDAM-DRW in [3]. -Mixed contributions: The authors' final results are based on Balanced softmax + Meta-resampling + decoupled training. Note that, decoupled training is compatible with most of the existing methods, and may lead to unfair comparison. The Meta-resampling is very similar to [12] and I would also view it as one trick that can be added on to all existing methods. Thus, the authors must show pure Balanced softmax to compare with existing methods for a fair comparison. -Overall, I like the proposed Balanced softmax and theoretical analysis. I particularly like the simplicity and cleanness of the method. However, other tricks that are added on to Balanced softmax make the contributions/claims a little vague. I do hope that the authors can focus on Balanced softmax. -Given these remaining concerns, I keep my score as 4.


Review 3

Summary and Contributions: The paper focuses on classification, in a long-tailed learning setup. I.e. where training class distribution is imbalanced. The paper main contribution is in identifying that using softmax is biased for training with class imbalanced dataset and testing with a balanced (uniformly distributed) dataset. the paper suggests a "balanced" variant for the softmax (which is different than re-weighting the loss function), and shows that this variant minimizes a respective generalization bound. In addition, is suggests a (meta-learned) class sampling scheme to further improve the empirical performance. Overall, the paper has nice theoretical and empirical results. I do have some criticism about some of the motivations and reported results and yet, I tend toward acceptance.

Strengths: +Useful analytical observation +Theoretical support (generalization bound) +Achieves equal/better SoTA performance on reported benchmarks

Weaknesses: (See additional details under my comments on "Correctness") -- Motivation for the additional (class) meta sampling is lacking. If the suggested softmax activation is the correct one, why would we need an additional sampling scheme? -- Similarly, why decoupled training is necessary? -- Ablation study and some baselines are missing. -- The quoted CIFAR results are difficult to compare with prior work. -- It will be beneficial to compare (both analytically, vs eq 4, and empirically) to a balanced loss reweighting scheme, where loss of a sample is reweighed by the class frequency at training time.

Correctness: The quoted CIFAR results are difficult to compare with prior work from multiple reasons: (1) The authors retrained (all?) the baselines themselves and at the same time report accuract while the convention for CIFAR is using error. (2) The baselines have mutiple variant, and it is unclear which one did the authors use for reproducing the results (e.g. is reproduced LDAM is just LDAM, or is it LDAM-DRW?) Both (1+2) make it hard verify the proposed approach compared to reported prior-work lines 145-149 and empirical results have contradicting claims: The claim that (1/4) exponent is optimal, yet conclude that the theoretical analysis lead to eq 4, which does not use the (1/4) exponent. Empirical results also show that (1/4) is not optimal. Line 156: the paper claims that CBS with Balanced-Softmax worsen performance, however their empirical results contradict that claim: Ablation study, compare row (6) with row (7). The introduction suggests a "multi-label" setup (line 29). However the paper only deals with the multi-class setup. Both analytically and empirically. Is it a typo? For the empirical part: (1) I would be thankful to see training with just the balanced-softmax, (without meta-scheduling) both with and without decoupled training, on all the datasets (not just for CIFAR). This will be beneficial to the community, in order to understand the gain in using the proposed balanced-softmax alone. (2) It would be beneficial to add comparison to a simple balanced loss scheme, on imagenet-LT or places-LT.

Clarity: Writing is clear.

Relation to Prior Work: Prior work is well cited, except for results on CIFAR, as detailed above.

Reproducibility: Yes

Additional Feedback: line 157-165 are better fitted to be a supplemental section. line 34: change "the generalization error bound" to "a generalization …" Post rebuttal: I have read the response and the other reviewer comments. Thank you for the clarifications.


Review 4

Summary and Contributions: The authors propose methods to train classifiers under large class imbalance. They show that vanilla cross-entropy is biased in such settings, and introduce a "Balanced Softmax" loss to address this, based on probabilistic arguments and arguments based on generalization error bounds. Moreover, they propose a strategy to learn an appropriate sampling of the training set under settings with very large class imbalances. They do extensive experiments on long-tailed classification and instance segmentation datasets, studying the interplay between the proposed loss and sampler.

Strengths: - The problem of long-tailed classification is very actual and relevant. - The paper offers both theoretical insights (probabilistic arguments and generalization bounds), and extensive experiments - The experimental results are strong and diverse (multiple datasets, imbalance factors, decoupled trainings or not...)

Weaknesses: - The 1/4 factor in the generalization bound is a bit unsatisfactory, although the authors do not try to shy away from this point and make experiments with both 1/4 and 1 factors.

Correctness: .The claims and method seem sound.

Clarity: The paper is well written and easy to follow.

Relation to Prior Work: Satisfactory discussions are done in terms of related works.

Reproducibility: Yes

Additional Feedback: Questions: 1. The authors mentioned that the learning of the sampling using the Gumbel-softmax trick for optimization is computationally costly. Could the authors expand on this question and explain the source of this cost, and how the approach scales in practice? Minor: - l.96 "different label distribution p(y = j|x)" -> I think this should be "p(y = j)" - Eq. (6) y subscript missing in l(θ) -> l_y(θ) - l.146-147 "We empirically find that setting 1/4 instead of 1 leads to the optimal results" -> shouldn't this be reversed (1 instead of 1/4?) - The paper mentions "meta sampler" and "meta reweighter" but the concept of meta reweighter is only explained in the appendix, I would recommend to add an explanation of meta reweighter in the main text for ease of reading. === post-rebuttal The concerns of fair comparison to existing methods in Table 2 raised by reviewer 2 are justified, and I am slightly downgrading my score from 8 to 7 in light of these comments. However, I still believe the paper has significant novelty, presents well-performing accuracies on established long-tail classification datasets, and is therefore worthy of acceptance.

[Author Response · NeurIPS 2020]

We thank all reviewers for encouraging our work on the following strengths: 1) Balanced Softmax is simple yet effective;
2) our theoretical analysis shows inspiring insights; 3) our experiments are extensive and performance achieves SOTA.
We will answer the major points below and address all remaining ones in the final version.
**Reviewer #1**:
**Q1**: Explanation about the mismatch (1/4 and 1) between the theory (Theorem 2 and Corollary 2.1) and practice.
**A1**: We used the slow rate $1/n^{\frac{1}{2}}$ in the derivation of Theorem 2 (see Sup. Mat.). [3] discussed that deep neural networks
can improve the convergence rate. When the convergence rate used in Theorem 2 is $1/n^2$, the factor in Corollary 2.1
will be 1 and aligns with Balanced Softmax. We leave further discussions on the convergence rate to future works.
**Reviewer #2**:
**Q1**: Eqn.3 and Eqn.4 are very similar to [3, A, B], ... particularly similar to Eqn.11 in [B].
**A1**: We progress the line of works [3, A, B] by introducing novel probabilistic insights that also bring significant
empirical improvements. Eqn.11 in [B] is generic (a superset of most loss engineerings like [3, 29, A]), it uses bi-level
optimization to find the unknown logit adjustment $\xi_{p,j}$ of each class, leaves a large search space and a hard optimization
landscape. We directly derive the optimal logit adjustment ($\xi_{p,j} = n_j$) with a solid probabilistic grounding (Theorem
1). Moreover, none of [3, A, B] touches the core observation of our work: the link between Softmax and the Bayesian
inference under data-imbalanced scenarios. We will add a discussion on [3, A, B] in the final version.
**Q2**: Meta sampler has a similar idea to [12,24,27].
**A2**: [12,24,27] 's idea is to use meta-learning to find each training sample's importance towards model training, while
we proposed Meta Sampler as a viable solution to the over-balance issue described in line 151-165. Moreover, none of
the existing works extend from reweight to resample (Meta Sampler outperforms Meta Reweighter by a large margin on
CIFAR10-LT); theirs are instance-based and ours is class-based (fewer parameters and simpler optimization landscape).
**Q3**: The analysis does not imply proposed softmax... adding the margin term into the loss won't affect the learning.
**A3**: We did not suggest to add a margin constant into the loss term, instead, we use Corollary 2.1 to show that the
optimal margin can be achieved by a proper loss parameterization, i.e., the 1/4 variant of Balanced Softmax.
**Q4**: The authors argued that *re-sampling techniques* can be harmful to model training, but finally still apply it.
**A4**: The argument is for *Class Balanced Sampling*, but not for all *re-sampling techniques* (line 151-165). Please kindly
refer to R3Q1 for why we need Meta Sampler as a learnable re-sampling technique to complement Balanced Softtmax.
**Q5**: When to start the meta sampler leads to a mother hyper-parameter.
**A5**: We apply the Meta Sampler from the very beginning of the training (epoch 0) like any other re-sampling strategy
(e.g., Class Balanced Sampling), thus when to start Meta Sampler is not a mother hyper-parameter in our method.
**Q6**: Meta Sampler makes the contributions vague; include experimental results w/ and w/o the Meta Sampler.
**A6**: Meta Sampler is complementary to Balanced Softmax (line 38-39), which can be supported by the ablations on
CIFAR-LT (Table 5). We provide more results on LVIS with only Balanced Softmax: $AP_m$:26.3, $AP_f$:28.8, $AP_c$:27.3,
$AP_r$:16.2, $AP_b$:27.0. Compared to experiments in Table 4, the results show that BALMS works better as a whole.
**Q7**: The authors' baseline softmax results are much higher than those reported in other papers.
**A7**: Our baseline softmax results align with the most recent paper [29] (Table 7, CIFAR-100-LT), which is published on
CVPR 2020. Please kindly refer to R3Q3 for why we retrain all compared methods on the baseline.
**Reviewer #3**:
**Q1**: Motivation for the additional (class) meta sampling is lacking.
**A1**: We need Meta Sampler to appropriately re-sample according to Balanced Softmax's effect on gradients. The
'over-balance' analysis shows a hypothesized case: when the training loss *infinitely approaches* 0 (line 160-162),
Balanced Softmax will cast an inverse weight $1/n_j$ to gradients (its combination with Class Balanced Sampler makes
the overall weight approach $1/n_j^2$, i.e., over-balanced). However, when the training loss does not *infinitely approach* 0
(in actual training), Balanced Softmax's effect on gradients can be viewed as variables between 1 and $1/n_j$. Therefore,
we need to explicitly estimate the optimal sample rate to keep the gradient always being balanced weighted at $1/n_j$.
**Q2**: Why decoupled training is necessary?
**A2**: Decoupled training is not necessary. We used the technique in our work to: 1) align with recent research results
([15] ICLR 2020, [33] CVPR 2020) to benefit future study, and to 2) save the computational cost of Meta Sampler.
**Q3**: The quoted CIFAR results are difficult to compare with prior work.
**A3**: We retrained all compared methods since prior works chose different baselines and cannot be fairly compared
with. We used the highest softmax baseline ([29], CVPR 2020), and it is more challenging and revealing to achieve
performance gain on a higher baseline. Following the suggestions, we will specify more details on baseline variants.
**Reviewer #4**:
**Q1**: The 1/4 factor in the generalization bound is a bit unsatisfactory.
**A1**: The mismatch can be reasonably explained. Please kindly refer to our discussion on convergence rates in R1Q1.
**Q2**: Could the authors explain the source of this cost (Meta Sampler), and how the approach scales in practice?
**A2**: Meta Sampler involves a second-order optimization, it usually doubles the computational graph and triples the
forward/backward times. Thus, end-to-end training with it is slower. In practice, with decoupled training, we only
optimize for the linear classifier, which greatly reduced the #parameters in the loop and makes the cost acceptable.

[Meta-Review · NeurIPS 2020]

The paper first shows that the softmax gives a biased gradient estimation under the long-tailed setup, and proposes a balanced softmax to accommodate the label distribution shift between training and testing. Theoretically, the authors derive the generalization bound for multiclass softmax regression. They then introduce a balanced meta-softmax procedure, using a complementary meta sampler to estimate the optimal class sample rate and further improve long-tailed learning.Experiments demonstrate that this outperforms SOTA long-tailed classification solutions on both visual recognition and instance segmentation tasks. The paper was reviewed by the four reviewers that found strengths and weaknesses. The strengths were the fact that the idea is intuitive and simple to implement, the theoretical derivations in support of the method, and the good results. The weaknesses include the fact that method combines several ideas and it is difficult to see exactly which most contributes to its success, concerns about the experimental results that may invalidate the significance of the SOTA claims (use of strong baselines), similarity to the proposed softmax to others in the literature, similarity of the meta sampler to others in the literature, lack of discussion of these issues, and many other questions of technical detail. The author rebuttal satisfied some of the concerns but not all, e.g. even the most positive reviewer acknowledged questions about the fairness of the experimental set-up. In the end, the paper had three positive reviews of relatively low confidence and one very confident negative review, which raised most of the weaknesses above.